# Probabilistic Forecasting of Residential Energy Consumption Based on SWT-QRTCN-ADSC-NLSTM Model

**Ning Jin [1], Linlin Song [1], Gabriel Jing Huang [2,3] and Ke Yan [1,3,4,*]**

1    Key Laboratory of Electromagnetic Wave Information Technology and Metrology of Zhejiang Province, College of Information Engineering, China Jiliang University, Hangzhou 310018, China
2    Institut Polytechnique de Paris, Rte de Saclay, 91120 Palaiseau, France; gabriel.jing.huang@gmail.com
3    Hangzhou Gisway Information Technology Co., Ltd., Hangzhou 311121, China
4    Department of the Built Environment, College of Design and Engineering, National University of Singapore, Singapore 117566, Singapore
*    Correspondence: keddiyan@gmail.com; Tel.: +65-9875-5205

**Abstract:** Residential electricity consumption forecasting plays a crucial role in the rational allocation of resources reducing energy waste and enhancing the grid-connected operation of power systems. Probabilistic forecasting can provide more comprehensive information for the decision-making and dispatching process by quantifying the uncertainty of electricity load. In this study, we propose a method based on stationary wavelet transform (SWT), quantile regression (QR), Bidirectional nested long short-term memory (BiNLSTM), and Depthwise separable convolution (DSC) combined with attention mechanism for electricity consumption probability prediction methods. First, the data sequence is decomposed using SWT to reduce the complexity of the sequence; then, the combined neural network model with attention is used to obtain the prediction values under different quantile conditions. Finally, the probability density curve of electricity consumption is obtained by combining kernel density estimation (KDE). The model was tested using historical demand-side data from five UK households to achieve energy consumption predictions 5 min in advance. It is demonstrated that the model can achieve both reliable probabilistic prediction and accurate deterministic prediction.

**Keywords:** electricity consumption probabilistic forecasting; quantile regression; depthwise separable convolution; nested long short-term memory

## 1. Introduction

Accurate prediction of electricity consumption is an indispensable condition for the stable operation of smart grid systems, which provides important decision-making support for the energy dispatching system and power market bidding system [1,2]. With the acceleration of digitalization and the development of communication technologies, the frequency of load collection by many smart meters is also increasing [3], providing a high-quality, quantitative data set for load characterization and load forecasting, which in turn provides a data basis for short-term energy consumption forecasting.

By realizing ultra-short-term home power prediction in 5-min steps, understanding the real-time electricity consumption of households can help people respond to emergency power accidents in time. At the same time, it can better balance the power load and improve the efficiency of real-time dispatch and energy management of renewable energy resources in the microgrid [4]. With a large number of smart homes in use, accurate forecasting of energy consumption demand has become an essential part of deploying an energy management system (EMS) [5]. Consumers can monitor their electricity usage in real-time and take energy-saving measures to reduce energy bills.

However, due to the lifestyle [6] of family members, changes in the use of electrical appliances [7], and climate [8], electricity consumption data is non-linear, highly volatile,

and random. Also, given the complex nature of household electricity and the errors in sensor data [9], traditional time series prediction techniques are difficult to meet the accuracy requirements. For the past few years, with the rise of deep learning (DL) techniques, DL models such as long short-term memory neural networks (LSTM) have been increasingly applied to the study of time-series data.

Probabilistic prediction of electricity consumption has become a hot topic in current research. This paper presents a probabilistic prediction model combining an improved SWT, DL models, and QR. The model first uses SWT to decompose the data into low-frequency and high-frequency components. The decomposed low-frequency components (the main signal) become smooth and easy to predict after removing noise, so it can be accurately predicted using a conventional deep learning model, considering that the temporal convolutional network (TCN) has powerful timing processing capability and its parallel computing, so its main trend information is predicted using TCN. The high-frequency part of the wavelet decomposition still has violent and frequent fluctuations. We will perform a secondary wavelet decomposition on the high-frequency fluctuation signal and then use a model with powerful feature extraction capability to make predictions. This study uses a hybrid model combining BiNLSTM, DSC, and attention, namely ADSC-NLSTM, to predict high-frequency sequences. These DL modules are combined with QR to directly generate quantile prediction results of load to derive prediction intervals. Finally, use kernel density estimation to generate probability density distribution. The main contributions of our study can be listed as follows:

- The original data is decomposed by SWT to generate multiple sub-signals, and its high-frequency signal is decomposed twice and trained using TCN and ADSC-NLSTM for both low and high-frequency signals. This combination helps to solve the model adaptation problem for electricity consumption prediction.
- The proposed ADSC-NLSTM network can make predictions in space and time. DSC based on the attention mechanism can preserve important information, prevent information loss and enhance feature extraction performance. The nested structure of the BiNLSTM is used to perform deep feature extraction effectively, and there are more recurrent units to obtain the dependencies of features at each time point, allowing the network to learn more adequately.
- This model is compared with existing frontier technologies, including individual models and combined models, and the experimental results show that this model achieves efficient energy consumption point prediction and probability prediction.

## 2. Related Works

Due to the compositional differences between family members and the unpredictability of human behavior, predicting household energy consumption becomes more challenging and a complex time-series prediction problem. In recent years various forecasting methods have been proposed to predict energy consumption, and these techniques can be classified as traditional models, machine learning (ML) models, and deep learning (DL) models [10].

Traditional methods mainly include the autoregressive moving average model (ARMA) method [11], the autoregressive integrated moving average (ARIMA) method [12], and some other statistical methods. These methods are more suitable for analyzing the linear part of the historical data and have poor prediction results for non-linear and complex data.

The decision tree [13], support vector machine (SVM) [14], random forest (RF) [15], and other machine learning methods have been widely used in the field of load prediction. Researchers have used machine learning to train many models to tune the prediction accuracy, such as Wei R R et al. [16] proposed a load forecasting method of principal component analysis (PCA) and the Least square-support vector machine (LS-SVM). T. Pinto et al. [17] proposed an integrated learning model with three machine learning algorithms: RF gradient augmented regression trees and Adaboost. However, current machine learning methods are prone to fall into local optima because of the challenging dynamic correlation between

variables and the change in data characteristics over time. When overfitting occurs, it is difficult to determine long-term and reliable use.

Deep learning, with its greater generalization ability and faster computational speed, has achieved significant results in feature extraction and learning of non-linear mapping relationships. The popular deep learning models currently include LSTM [18], convolutional neural network (CNN) [19], TCN [20], gate recurrent unit (GRU) [21], Generative Adversarial Network (GAN) [22], and other neural network models. These methods have been widely used in various fields. Kong W et al. [23] proposed an LSTM-based model for short-term residential load forecasting, which gives their model an advantage over state-of-the-art machine learning models and empirical models.

Currently, any single DL model has some limitations for predicting energy consumption time series data with many mutation values. To further improve the accuracy of electricity consumption prediction, many scholars have proposed hybrid models that combine data decomposition techniques with prediction models. The common data decomposition methods include empirical mode decomposition (EMD) [24], wavelet transforms [25], variational modal decomposition (VMD) [26], and so on. Researchers usually use data decomposition techniques to decompose the original power energy sequence into a finite number of components with mutually different characteristics and then build prediction models for each component separately or use each component as an input feature of a single prediction model. Cai et al. [27] decomposed the power load sequence data into modal components of different frequencies by VMD. They used GRU and TCN to predict the low-frequency and high-frequency components, respectively. Inspired by this, we make predictions for the components of different frequencies using the corresponding model. Shao et al. [28] used discrete wavelet transform (DWT) reconstructed sequences as the input for power consumption prediction. Then they built a combined prediction model based on CNN-LSTM for each subseries. However, like most wavelet decomposition methods, this article only focuses on the decomposition of low-frequency approximate signals, not high-frequency signals. In view of this problem, Zeng Y et al. [29] proposed a new model combining ESWT and NSTM and performed well in the air quality prediction effect. Liang Y. et al. [30] combined ICEEMDAN, LSTM, CNN, and CBAM to create a collaboration that has built a powerful model. This joint approach can take full advantage of the respective algorithms' strengths to improve the model's performance and robustness. However, the ICEEMDAN algorithm has a long computation time, which is not conducive to short-term prediction.

The literature mentioned above basically provides only a single-point prediction result, and there are relatively few studies on uncertainty prediction. Realizing the interval and probability predictions of energy consumption can provide more information [31]. The quantile regression [32] method can be combined with other methods, such as neural networks, without prior distribution assumptions, and has been widely used in time-series probability forecasting. Wang et al. [33] used the quantile loss function to guide the training process and applied the traditional LSTM network in quantile to load-interval probability forecasting.

## 3. Methodology

This section presents our proposed hybrid deep learning prediction framework for electricity consumption prediction. We normalize the data for pre-processing and then decompose the univariate data into several high-frequency and low-frequency components by SWT, effectively focusing on all signal details. The low-frequency components are then used as input to the TCN, and the high-frequency components are used as input to the DSC-NLSTM. Then the final prediction results are output via a fully connected layer. The overall structure of the proposed method is shown in Figure 1.

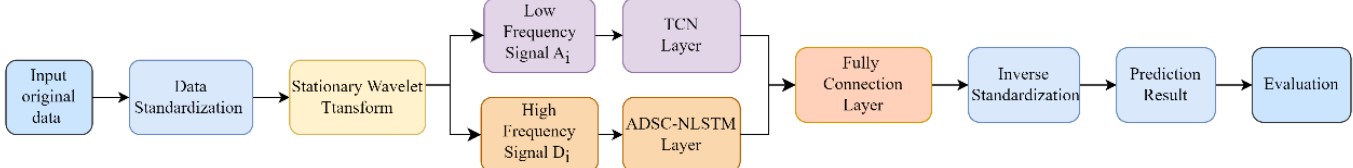

**Figure 1.** The overall structure of the proposed method.

*3.1. The Stationary Wavelet Transform*

The noisy and highly variable characteristics of the original residential electricity consumption time series data make its features difficult to be captured by deep learning models. The wavelet transform is a transform analysis method that replaces the Fourier decomposition of the sine and cosine waves with a set of orthogonal bases that decay with time, so it can obtain both time-domain and frequency-domain information and better represent the abrupt and non-smooth parts of the series.

The standard DWT uses a binary extraction algorithm to downsample the signal, which reduces the wavelet coefficients by half after each decomposition, thus losing detailed information about the original signal with each decomposition. The SWT, which removes downsampling while upsampling the filter, solves this problem well and maintains the full waveform better than the DWT [34].

The original signal $Y(t)$ after the k-level SWT decomposition can be expressed as

$$Y(t) = A_{kt} + \sum_{i=1}^{k} D_{it} \tag{1}$$

where $A_{kt}$ denotes the approximate component of the signal obtained through the low-pass filter, representing the general trend of the time series, and $D_{it}$ denotes the detailed component of the signal obtained through the high-pass filter, representing the local complex fluctuations caused by various transient factors. $A_{kt}$ is further decomposed into a high-frequency signal $A_{(k+1)t}$ and a low-frequency signal $D_{(k+1)t}$.

Conventional SWT only further decomposes the low-frequency signal $A_k$ in the process of decomposition, and no further decomposition is implemented for the high-frequency signal. Therefore the high-frequency sub-series D1 generated by the first wavelet decomposition contains the highest frequency noise and the most intense local fluctuations in the original data, which is more difficult to predict. We will perform a second wavelet decomposition of D1 so that the time series features within the high-frequency subcomponent can be more effectively and accurately expressed, with more accurate local analysis capability.

In this study, Daubechies wavelet is chosen as the basis function, and the number of decomposition layers was set to 3. As shown in Figure 2, the original data is decomposed into A3, D1, D2, and D3 by wavelet decomposition, and then the D1 signal is further decomposed to obtain D1-A1, D1-D1. Finally, we get A3, D3, D2 and D1-A1, D1-D1, a total of 5 sub-series.

*3.2. Deep Learning Module*

This study uses deep learning models to extract features from the decomposed sequence data.

3.2.1. Bidirectional Nested Long Short-Term Memory

A nested LSTM neural network (NLSTM) is a transformed form of LSTM that increases the depth of the LSTM by nesting rather than stacking [35]. The incoming information in the NLSTM adds a recursive external storage unit inside the network unit that can selectively read the long-term information learned by the internal unit. Overall, the robustness of the original LSTM neural network structure has been improved. The structure of the NLSTM is shown in the left part of Figure 3.

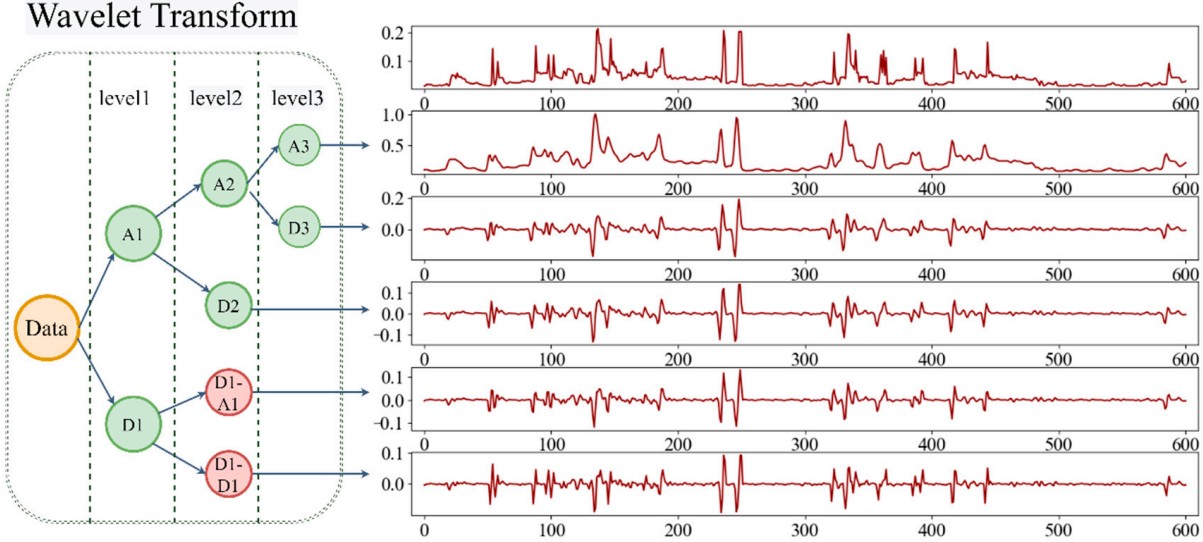

**Figure 2.** Decomposition component after wavelet transform. The graphs on the right represent the raw data from top to bottom, A3, D3, D2, D1-A1, and D1-D1. Only 600 data samples were depicted for clearer vision.

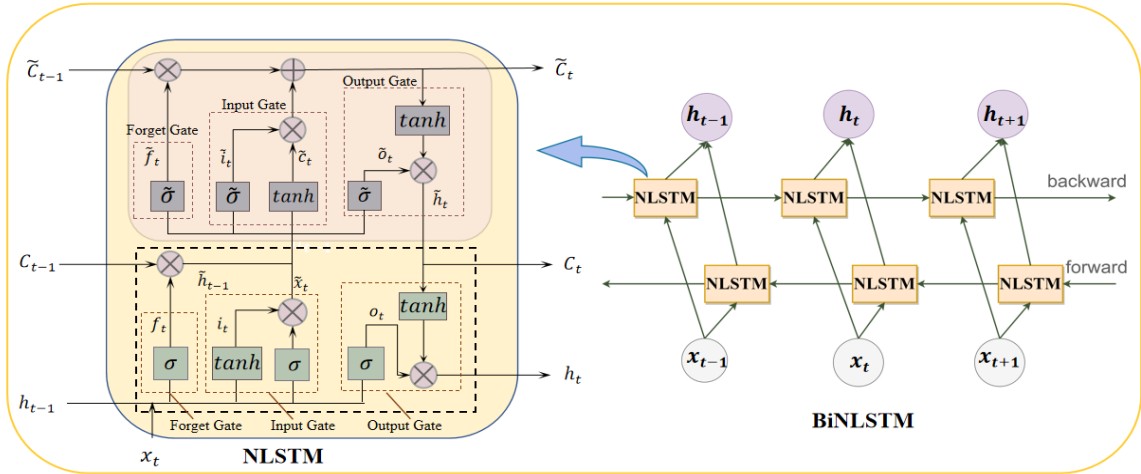

**Figure 3.** The structure of NLSTM and BiNLSTM memory cells. The pink shaded part in the left picture is the inner LSTM unit of NLSTM.

The equations that update the cell state and gates in an inner LSTM unit are similar to a common LSTM memory cell, as shown as follows:

$$\widetilde{f}_t = \sigma\left(\widetilde{W}_f \cdot \left[\widetilde{h}_{t-1}, \widetilde{x}_t\right] + \widetilde{b}_f\right) \tag{2}$$

$$\widetilde{i}_t = \sigma\left(\widetilde{W}_i \cdot \left[\widetilde{h}_{t-1}, \widetilde{x}_t\right] + \widetilde{b}_i\right) \tag{3}$$

$$\widetilde{o}_t = \sigma\left(\widetilde{W}_o \cdot \left[\widetilde{h}_{t-1}, \widetilde{x}_t\right] + \widetilde{b}_o\right) \tag{4}$$

$$\widetilde{C}_t = \widetilde{f}_t \odot \widetilde{C}_{t-1} + \widetilde{i}_t \odot \widetilde{c}_t \tag{5}$$

$$\widetilde{h}_t = \widetilde{o}_t \odot tanh\left(\widetilde{C}_t\right) \tag{6}$$

where $\sigma$ is sigmoid activation function; $\widetilde{W}_f$, $\widetilde{W}_i$, $\widetilde{W}_o$, and $\widetilde{W}_c$ are the weight matrices connected to the forget gate $\widetilde{f}_t$, the input gate $\widetilde{i}_t$, the output gate $\widetilde{o}_t$, and the memory cell $\widetilde{C}_t$, respectively; $\widetilde{b}_f$, $\widetilde{b}_i$, $\widetilde{b}_o$, and $\widetilde{b}_c$ are the corresponding biases. $\widetilde{h}_{t-1}$ and $\widetilde{x}_t$ are the input to the inner LSTM unit and are calculated based on the parameters of the external unit, which are given by the calculation formula:

$$\widetilde{h}_{t-1} = f_t \odot C_{t-1} \tag{7}$$

$$\widetilde{x}_t = i_t \odot tanh(W_c \cdot [x_t, h_{t-1}] + b_c) \tag{8}$$

When the internal memory cell has finished calculating, the external memory cell state $C_t$ is updated as follows:

$$C_t = \widetilde{h}_t \tag{9}$$

The NLSTM captures long-term dependencies from front to back and can only capture past features. While as shown in Figure 3, BiNLSTM extracts the forward and backward relationships of the input data sequence, further improving the global and complete feature extraction with better prediction results. The output of the hidden layer is represented as follows:

$$h_t = h_{tf} \oplus h_{tb} \tag{10}$$

where $h_t$ is the hidden state vector of the BiNLSTM, $\oplus$ is the vector splicing calculation; $h_{tf}$ and $h_{tb}$ are the hidden state vectors of the forward NLSTM and backward NLSTM, respectively.

### 3.2.2. Temporal Convolutional Network

TCN is a novel model that can be used to solve time series prediction problems. As can be seen from Figure 4, TCN uses a one-dimensional convolutional network consisting of dilated causal convolution and residual blocks. It has the advantages of more stable gradients, higher computational efficiency, longer memory sequences, etc.

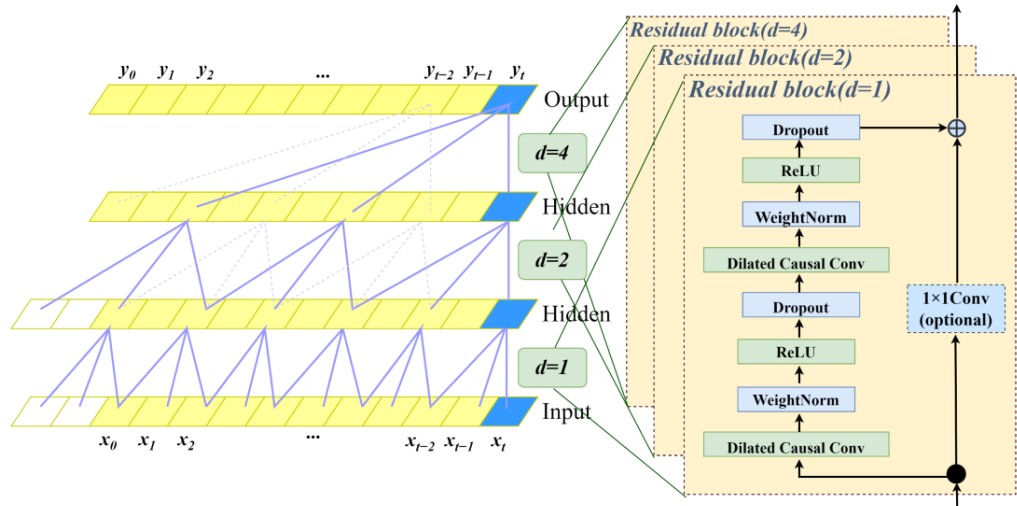

**Figure 4.** The structure of TCN. The graph on the right represents the dilated causal convolution, and the left shows the residual block.

Dilated causal convolution consists of causal convolution and dilated convolution [36]. Causal convolution is a strict time-constrained model that prevents future data from leaking into past data. Dilated convolution samples input at intervals on the basis of causal convolution. It adjusts the size of the receptive field by changing the expansion coefficient, which enables the network to flexibly adjust the amount of historical information

received by the output. The residual links in the residual block enable the network to pass information across the number of layers, thus avoiding the loss of information caused by too many layers.

The TCN network used in this paper uses a dilated causal convolution with expansion coefficients $d = 1, 2, 4$ and filter coefficients $k = 3$.

### 3.2.3. Depthwise Separable Convolution and Attention Mechanisms

Depthwise separable convolution is an improved algorithm for traditional convolution, which reduces the number of parameters required for convolutional operations by splitting the correlation between spatial and channel dimensions and is widely used for structural optimization of deep convolutional networks. The operation process of DSC is divided into two parts: depthwise convolution and point size convolution, and its structure is shown in Figure 5.

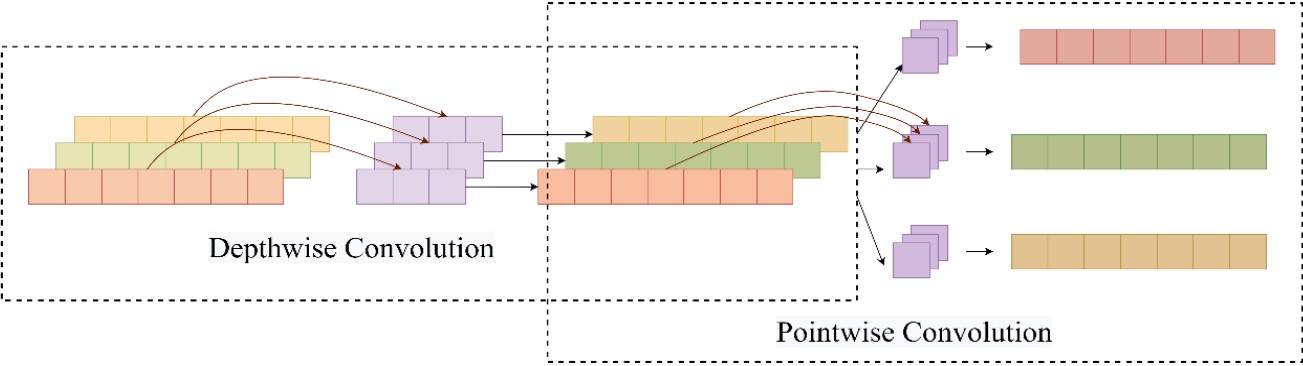

**Figure 5.** The structure of DSC.

The convolutional block attention module (CBAM) is one of the more popular attention mechanisms in image processing [37]. Traditional convolutional neural networks use the same convolution for all feature channels, ignoring the different levels of importance of the information contained in the different channels, resulting in lower training accuracy. CBAM makes the network performance better by using channel attention module (CAM) and spatial attention module (SAM) in both channel and spatial dimensions, respectively. The internal structure of CBAM is shown in Figure 6.

CAM is expressed as follows:

$$
\begin{aligned}
F^c &= \sigma(MLP(AvgPool(F)) + MLP(MaxPool(F))) \\
&= \sigma\left(W_1\left(W_0\left(F^c_{avg}\right)\right)\right) + W_1(W_0(F^c_{max}))
\end{aligned}
\tag{11}
$$

where $\sigma$ is the sigmoid activation function; $MLP$ is the fully connected layer.

First, average pooling and maximum pooling are performed on the input feature map $F$ by channel, and then the two one-dimensional vectors after pooling are sent to the fully connected layer operation and added to form channel attention $F^c$. SAM is expressed as follows:

$$
F' = F \bigotimes F^c
\tag{12}
$$

$$
F^t = \sigma\left(f^{7\times7}\left(\left[AvgPool(F'); MaxPool(F')\right]\right)\right)
\tag{13}
$$

where $\bigotimes$ represents the multiplication of corresponding elements. $\sigma$ is the sigmoid activation function.

Then, the feature map $F'$ is obtained by multiplying the channel attention and the input elements, and $F'$ is spatially subjected to average pooling and maximum pooling. The two vectors generated by pooling are concatenated and then convolved to finally generate spatial attention $F^t$.

The final CBAM output is shown as follows:

$$F'' = F' \bigotimes F^t \tag{14}$$

where $\bigotimes$ represents the multiplication of corresponding elements.

Finally, multiply the spatial attention and $F'$ element-wise to get the final output feature $F''$.

In this paper, DSC is combined with CBAM, namely ADSC, where SAM allows the DSC network to focus more on the mutation regions in the data sequence that play a decisive role in the prediction effect and ignore unimportant information, while CAM considers the relationship between feature map channels. After two attention mechanisms, the prediction effect of the model will be improved.

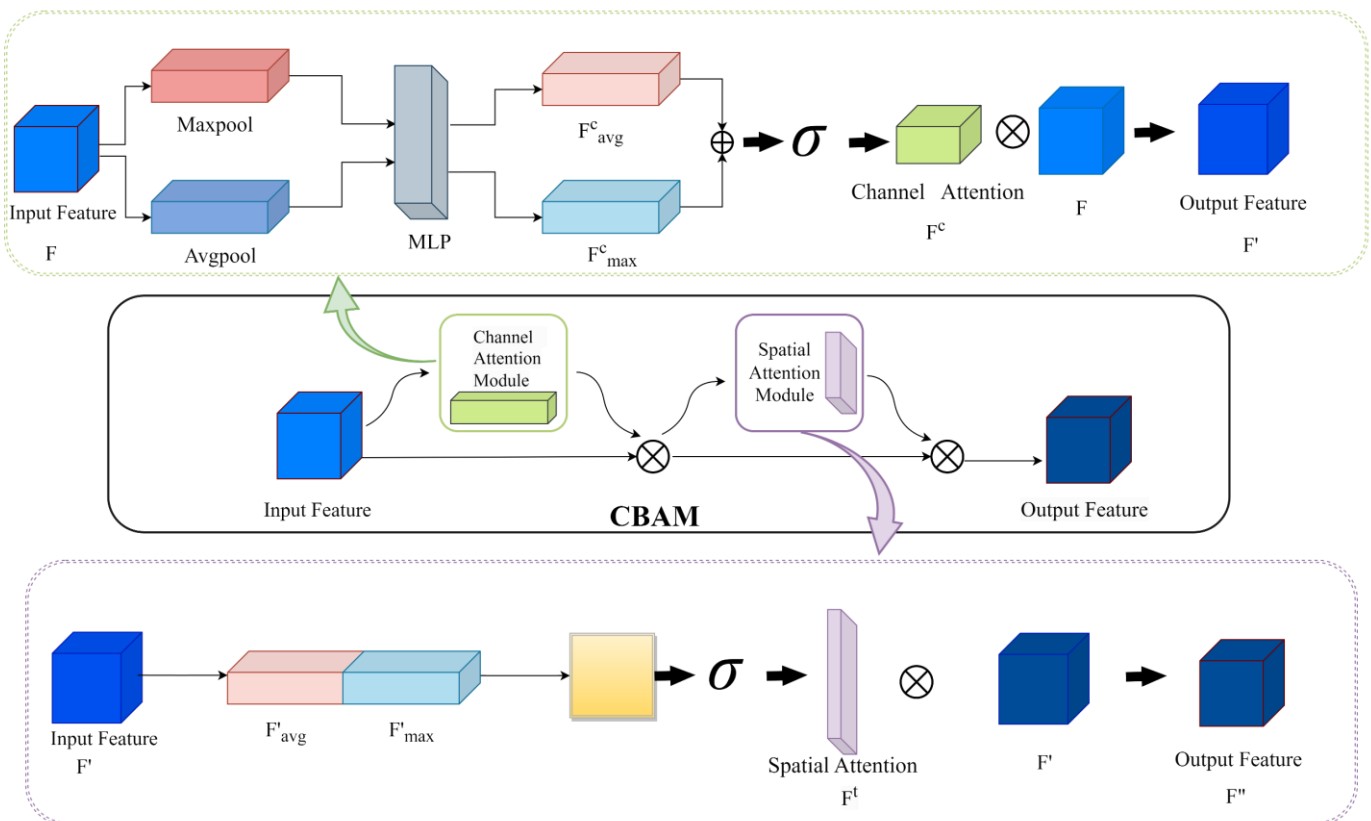

**Figure 6.** The structure of CBAM. The middle part of the figure is the overall structure of CABM, the top is the internal structure of CAM, and the bottom is the internal structure of SAM.

*3.3. Quantile Regression*

Quantile regression is a linear model that describes the relationship between the conditional quantile of the independent variables $X = [x_1, x_2, \ldots, x_n], x_i = [x_{i1}, x_{i2}, \ldots, x_{im}]$ and dependent variables $y = [y_1, y_2, \ldots, y_n]$ [38]. The linear QR model is as follows:

$$Q_{y_i}(\tau|x_i) = f(x_i, \alpha(\tau))$$
$$= \alpha_0(\tau) + \alpha_1(\tau)x_{i1} + \alpha_2(\tau)x_{i2} + \ldots + \alpha_n(\tau)x_{im} \tag{15}$$

where $\tau \epsilon (0, 1)$ is the quantile value; $Q_{y_i}(\tau|x_i)$ is the estimate of $y$ at the $\tau - th$ quantile; $\alpha(\tau) = [\alpha_0(\tau), \alpha_1(\tau), \ldots, \alpha_n(\tau)]^T$ are the regression coefficients at the $\tau - th$ quantile. The estimated value $\hat{\alpha}(\tau)$ of $\alpha(\tau)$ can be calculated by minimizing the loss function:

$$\hat{\alpha}(\tau) = argmin \sum_{i=1}^{n} \rho_\tau(y_i - f(x_i, \alpha(\tau))) \tag{16}$$

where $\hat{\alpha}(\tau)$ denotes the value at which the loss function is taken to be minimal, $\rho_\tau(u)$ is an asymmetric function based on quantile $\tau$ whose formula is as follows:

$$\rho_\tau(u) = \begin{cases} \tau u & u \geq 0 \\ (\tau - 1)u & u < 0 \end{cases} \tag{17}$$

When $\tau = 0.5$, the quantile loss function is equivalent to the mean absolute error. When $\tau < 0.5$, the loss function is more concerned with small predicted values, and when $\tau > 0.5$, the loss function is more concerned with large predicted values. Therefore, the quantile loss function can better adapt to the distribution of the data at different quartiles, allowing the quantile regression to more accurately predict the conditional probability distribution of the dependent variable.

*3.4. Kernel Density Estimation*

This paper uses kernel density estimation, which is a nonparametric estimation method without prior assumptions, and calculates the probability density function (PDF) curve by combining the output results of quantile regression [39]. If the predicted value of point y at each quantile is $\hat{y} = [\hat{y}_1 \hat{y}_2, \cdots, \hat{y}_n]$, the probability density function formula at $y$ is as follows:

$$p(y) = \frac{1}{nB} \sum_{i=1}^{n} K\left(\frac{\hat{y}_i - y}{B}\right) \tag{18}$$

where $B > 0$ is the bandwidth. $K(\cdot)$ is a kernel function. This study employs the Gaussian kernel [40] as kernel function. The formula can be expressed as follows:

$$K(\mu) = \frac{1}{\sqrt{2\pi}} exp\left(-\frac{\mu^2}{2}\right) \tag{19}$$

where $\mu = \frac{\hat{y}_i - y}{B}$.

*3.5. Combination Model of Electricity Consumption Probability Prediction*

As shown in Figure 7, the study proposes a novel electricity consumption probabilistic prediction model combining SWT, QR, TCN, BiNLSTM, DSC, and CBAM (SWT-QRTCN-ADSC-NLSTM). The original electricity consumption data is decomposed by a double wavelet to obtain a low-frequency component and four high-frequency components.

According to the different frequency characteristics of the decomposed subsequence, selecting different DL models for data training can better capture the different change patterns and trends of temporal signals. The low-frequency main signal becomes smooth after removing high-frequency noise, and a single normal model can capture its long-term trend. If complex models are used with too many training parameters, they not only increase the training time but may also lead to over-fitting. TCN can effectively grasp the change law of stationary signal in timing under the case of a few model parameters. This method is applied to predict low-frequency components. In contrast, the high-frequency signal obtained from the decomposition still contains both frequent and complex fluctuations, and it is difficult for a single model to capture the complex non-linear features. Therefore, a combined model with high robustness, ADSC-NLSTM, is proposed and applied to the high-frequency section.

The ADSC-NLSTM model is composed of two parallel networks, BiNLSTM and ADSC-CBAM, which capture the characteristics of data from time and space, respectively. Among them, CBAM weights the feature map obtained by the DSC model, highlights the feature information in the feature map that has a greater effect on the prediction performance and improves the prediction ability of high-frequency signals. The multidimensional feature data obtained from the TCN-ADSC-NLSTM model after extracting the high and low-frequency components are fed into the fully connected layer, and then the quantile

regression is used, i.e., the loss function of the prediction model is set to a quantile loss function according to Equation (16).

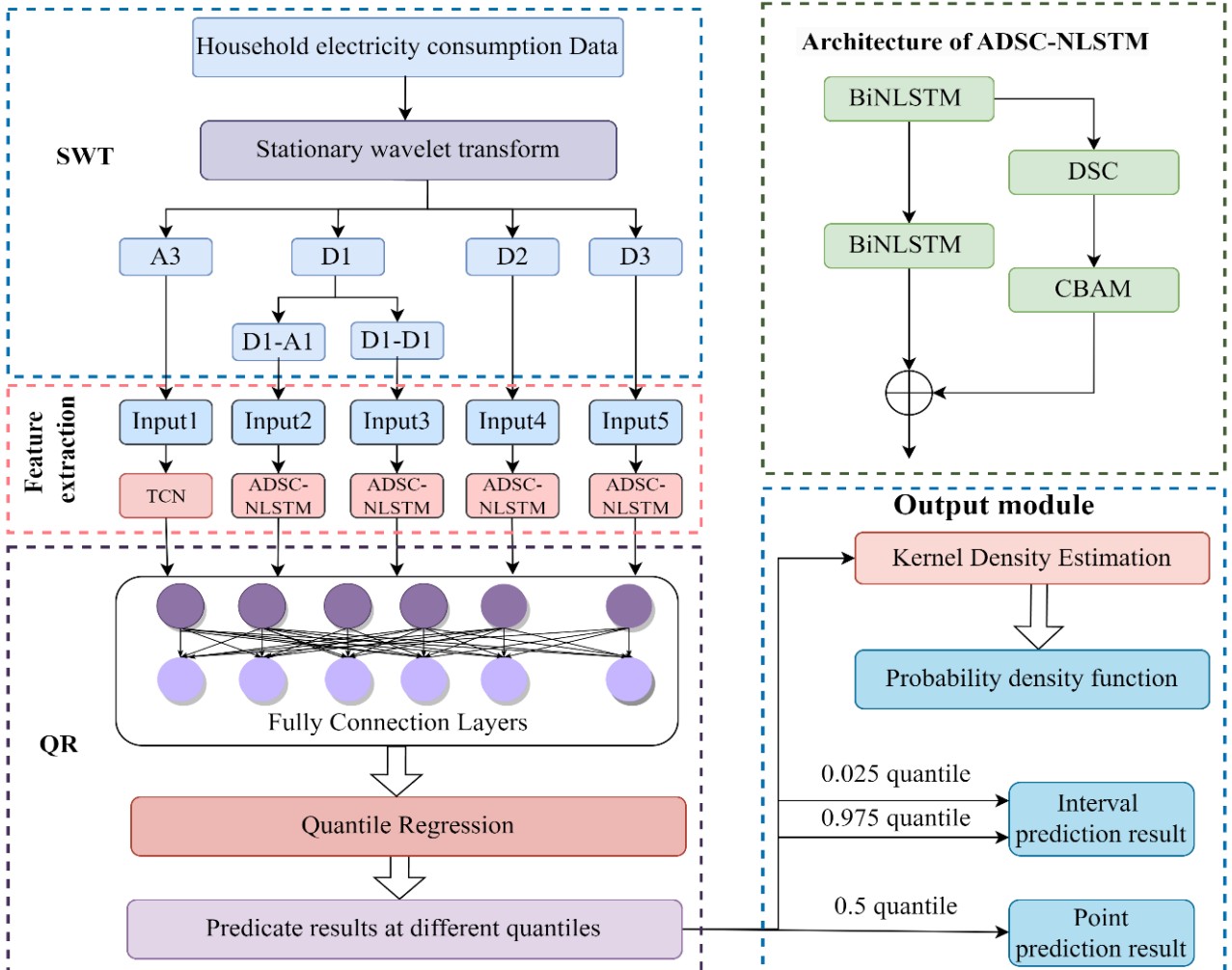

**Figure 7.** The frame diagram of the residential electricity probabilistic forecasting method based on quantile regression and deep learning module.

In this paper, a total of 99 quantile points with an interval of 0.01 are selected from 0.01 to 0.99 quantile points $\tau = [0.01, 0.02, \ldots, 0.98, 0.99]$. When $\tau = 0.5$, it is used as a point prediction result. When $\tau = [0.025, 0.975]$, it is used as an interval prediction result with a confidence level of 95%. The probability prediction result is obtained by calculating the 99 quantile prediction results through KDE.

## 4. Experimental Process and Results

### 4.1. Dataset Description

The open-source data we use comes from a project called "UK-DALE" [41]. This dataset records power consumption data for each appliance and five households' total electricity consumption. The entire dataset spans from 22:28:15 on 9 November 2012 to 18:35:53 on 26 April 2017, where the sampling interval is 6 s. We take the cumulative sum of 50 consecutive sampling points each time to obtain the 5-min interval data. We selected the first 9600 data samples for each household as the amount of data varies between households. The electricity consumption data of each household was divided into a training set and a test set according to the ratio of 87.5% and 12.5%.

*4.2. Evaluation Metrics*

The probabilistic prediction model designed in this paper provides point prediction, interval prediction, and probability density estimation. In order to verify the validity of forecasts, we have used the corresponding evaluation metrics to verify the validity of the forecasts.

4.2.1. Evaluation Metrics of Point Prediction

In order to assess the accuracy of the experimental predictions, three evaluation metrics were used, including mean absolute error (MAE), root mean square error (RMSE), and mean absolute percentage error (MAPE). The calculation formula of each evaluation index MAE, RMSE, and MAPE is as follows:

$$\text{MAE}(y, \hat{y}) = \frac{1}{n} \sum_{i=1}^{n} |y_i - \hat{y}_i| \tag{20}$$

$$\text{RMSE}(y, \hat{y}) = \sqrt{\frac{1}{n} \sum_{i=1}^{n} (|y_i - \hat{y}_i|)^2} \tag{21}$$

$$\text{MAPE}(y, \hat{y}) = \frac{100\%}{n} \sum_{i=1}^{n} \left| \frac{y_i - \hat{y}_i}{y_i} \right| \tag{22}$$

where $n$ is the total number of samples, is the actual data value, and is the predicted value. MAE, RMSE, and MAPE are used to assess the deviation between the predicted and true values. The smaller these metrics are, the higher the point prediction accuracy.

4.2.2. Evaluation Metrics of Interval Prediction

In this paper, the prediction interval coverage probability (PICP) is used to calculate the probability of the actual value falling within the prediction interval and to describe the accuracy of the interval prediction. The prediction interval average width (PIAW) is also used to describe the magnitude of the fluctuation of the interval prediction. The above formula is as follows.

$$PICP = \frac{1}{n} \sum_{i=1}^{n} S_i \tag{23}$$

$$PIAW = \frac{1}{n} \sum_{i=1}^{n} \frac{y_{iup} - y_{idown}}{y_i} \tag{24}$$

where $n$ is the total number, is a Boolean function, and takes 1 when the observation falls within the prediction interval, otherwise takes 0. and the upper and lower bounds of the prediction interval, respectively. The larger PICP indicates the higher reliability of interval prediction. The smaller PIAW indicates that the average width of the prediction intervals obtained by the model is narrower, and the model has a higher interval prediction acuity.

4.2.3. Evaluation Metrics of Probability Prediction

The continuous ranking probability score (CRPS) was used to evaluate the probabilistic prediction performance of the model. The smaller the CRPS value, the better the probabilistic prediction performance of the model. The specific formula is as follows:

$$CRPS = \frac{1}{n} \sum_{i=1}^{n} \int_{-\infty}^{+\infty} [F(y_i) - I(\hat{y}_i - y_i)]^2 dy_i \tag{25}$$

$$F(y_i) = \int_{-\infty}^{y_i} p(x_i) dx_i \tag{26}$$

where $F(\cdot)$ is the cumulative density function; $x_i$ is the $i$-th sample input; $p(\cdot)$ is the probability density function; $I$ is the Heaviside function. The smaller the CPS, the better the overall performance and reliability of the model's probabilistic prediction.

### 4.3. Prediction Results Analysis

To evaluate the predictive performance of the hybrid model proposed in this paper, eight prediction models were selected as comparison models. These models are QRLSTM, QRGRU, QRNLSTM, QRCNN-LSTM, SWT-QRLSTM, SWT-QRNLSTM, SWT-QRDSC-NLSTM, SWT-QRADSC-NLSTM. Among them, SWT-QRDSC-NLSTM and SWT-QRADSC-NLSTM use the same decomposition as in this paper, and then the decomposed components are input to the parallel model for training, which belongs to the ablation study of this study and is used to verify the effectiveness of DSC and CBAM's for prediction effect.

#### 4.3.1. Point Prediction Results Analysis

The results of the point prediction evaluation metrics for the various models in the Family 1 test set are shown in Table 1, where the MAE, RMSE, and MAPE of the proposed algorithm are significantly reduced compared to the comparison models.

**Table 1.** Results of point prediction evaluation of the different models.

| | MAE | | | | | RMSE | | | | | MAPE (100%) | | | | |
|---|---|---|---|---|---|---|---|---|---|---|---|---|---|---|---|
| | Hse1 | Hse2 | Hse3 | Hse4 | Hse5 | Hse1 | Hse2 | Hse3 | Hse4 | Hse5 | Hse1 | Hse2 | Hse3 | Hse4 | Hse5 |
| QRLSTM | 0.0103 | 0.0101 | 0.0127 | 0.0098 | 0.0081 | 0.0223 | 0.0236 | 0.0328 | 0.0211 | 0.0245 | 22.31 | 24.71 | 29.32 | 28.12 | 9.21 |
| QRGRU | 0.0108 | 0.0099 | 0.0119 | 0.0098 | 0.0085 | 0.0226 | 0.0242 | 0.0320 | 0.0214 | 0.0235 | 23.44 | 21.38 | 23.77 | 27.62 | 10.39 |
| QRNLSTM | 0.0098 | 0.0106 | 0.0137 | 0.0091 | 0.0103 | 0.0218 | 0.0231 | 0.0315 | 0.0208 | 0.0232 | 20.03 | 27.64 | 34.29 | 22.03 | 15.51 |
| QRCNN-LSTM | 0.0099 | 0.0093 | 0.0130 | 0.0098 | 0.0077 | 0.0227 | 0.0238 | 0.0317 | 0.0217 | 0.0234 | 18.55 | 19.71 | 29.14 | 25.67 | 8.65 |
| SWT-QRLSTM | 0.0073 | 0.0041 | 0.0061 | 0.0022 | 0.0035 | 0.0144 | 0.0059 | 0.0108 | 0.0028 | 0.0040 | 16.26 | 12.20 | 13.54 | 11.31 | 6.99 |
| SWT-QRNLSTM | 0.0087 | 0.0037 | 0.0033 | 0.0026 | 0.0042 | 0.0154 | 0.0054 | 0.0042 | 0.0029 | 0.0070 | 17.92 | 11.47 | 14.71 | 13.43 | 6.35 |
| SWT-QRDSC-NLSTM | 0.0018 | 0.0017 | 0.0018 | 0.0026 | 0.0026 | 0.0035 | 0.002 | 0.0020 | 0.0029 | 0.0031 | 4.57 | 6.15 | 8.95 | 9.97 | 4.48 |
| SWT-QRADSC-NLSTM | 0.0011 | 0.0009 | 0.0011 | 0.0036 | 0.0021 | 0.0027 | 0.0023 | 0.0016 | 0.0077 | 0.0058 | 2.99 | 3.29 | 4.13 | 9.57 | 2.09 |
| Proposed | 0.0003 | 0.0003 | 0.0009 | 0.0013 | 0.0007 | 0.0006 | 0.0006 | 0.0014 | 0.0025 | 0.0012 | 0.91 | 1.02 | 2.96 | 3.54 | 1.05 |

SWT-QRDSC-NLST, SWT-QRADSC-NLSTM, and our proposed method were decomposed quadratically. Compared to SWT-QRLSTM and SWT-QRNLSTM, the errors in the prediction results of those former models are all reduced to different degrees. The decline is due to the fact that we re-decompose the D1 subseries of the most violent fluctuations. Adequate treatment of the high-frequency components allows for a more efficient and accurate representation of the time series features within the high-frequency subseries. At the same time, the effectiveness of DSC and CBAM in feature extraction was verified in the ablation experiment, including SWT-QRDSC-NLST, SWT-QRADSC-NLSTM, and SWT-QRTCN-ADSC-NLSTM, which played a certain role in the accuracy of point prediction result.

For a more intuitive representation of the prediction results, we show in Figure 8 the prediction curves of electricity consumption obtained with various models trained. The red line in the figure indicates the point prediction forecasts of the model proposed in this paper, while the black line indicates the actual energy consumption data.

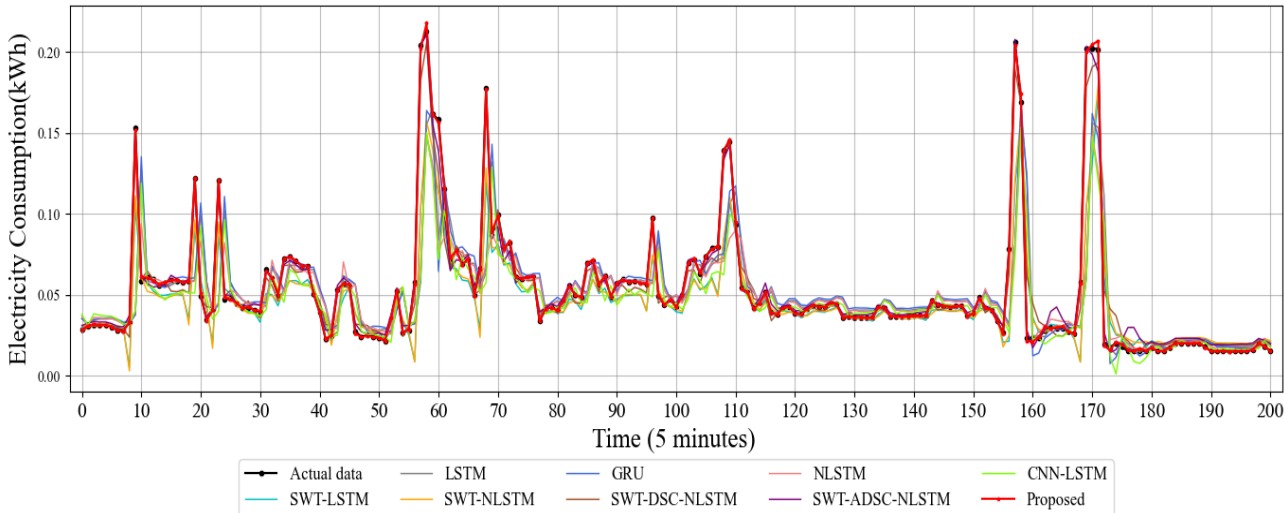

**Figure 8.** Point prediction results of different models.

The trend of the curves in the figure shows that the energy consumption data has irregular fluctuations. Single DL models, including LSTM, GRU, and NLSTM, are poorly fitted and suffer significant lag. The combined model based on SWT decomposition generally fits the actual value curve better, indicating that the data decomposition has a significant improvement in the prediction effect. Among all the compared methods, the method proposed in this experiment has a perfect fit for the actual values and has the highest accuracy of the prediction results. At the same time, the effectiveness of DSC and CBAM in feature extraction is verified in the ablation experiment, which plays a particular role in the accuracy of point prediction results.

### 4.3.2. Interval Prediction Results Analysis

To verify the effectiveness of the interval prediction results, the results of the interval prediction evaluation indexes of the nine models at a confidence level of 95% are shown in Table 2. And it can be seen from the table that the interval coverage of the model proposed in this study is the highest compared with other models and has high reliability.

**Table 2.** Results of interval and probability prediction evaluation of different models.

|  | PICP | | | | | PIAW | | | | | CRPS | | | | |
|---|---|---|---|---|---|---|---|---|---|---|---|---|---|---|---|
|  | Hse1 | Hse2 | Hse3 | Hse4 | Hse5 | Hse1 | Hse2 | Hse3 | Hse4 | Hse5 | Hse1 | Hse2 | Hse3 | Hse4 | Hse5 |
| QRLSTM | 0.964 | 0.965 | 0.937 | 0.859 | 0.901 | 2.142 | 1.844 | 1.589 | 1.563 | 1.341 | 0.0074 | 0.0303 | 0.0097 | 0.0071 | 0.0058 |
| QRGRU | 0.956 | 0.974 | 0.924 | 0.947 | 0.936 | 1.965 | 2.193 | 1.566 | 1.921 | 1.292 | 0.0075 | 0.0304 | 0.0098 | 0.0071 | 0.0059 |
| QRNLSTM | 0.946 | 0.929 | 0.951 | 0.952 | 0.945 | 2.051 | 1.802 | 1.268 | 1.652 | 1.139 | 0.0069 | 0.0281 | 0.0099 | 0.0076 | 0.0056 |
| QRCNN-LSTM | 0.961 | 0.974 | 0.936 | 0.946 | 0.950 | 2.058 | 1.751 | 1.705 | 2.544 | 1.157 | 0.0064 | 0.0243 | 0.0100 | 0.0070 | 0.0057 |
| SWT-QRLSTM | 0.945 | 0.952 | 0.945 | 0.964 | 0.956 | 1.812 | 1.481 | 1.310 | 1.417 | 0.91 | 0.0049 | 0.0145 | 0.0064 | 0.0032 | 0.0027 |
| SWT-QRNLSTM | 0.981 | 0.962 | 0.955 | 0.965 | 0.951 | 1.557 | 1.590 | 1.272 | 1.526 | 0.901 | 0.0056 | 0.0143 | 0.0056 | 0.0043 | 0.0024 |
| SWT-QRDSC-NLSTM | 0.977 | 0.965 | 0.945 | 0.955 | 9.572 | 1.113 | 1.213 | 1.210 | 1.311 | 0.838 | 0.0026 | 0.0009 | 0.0018 | 0.0010 | 0.0016 |
| SWT-QRADSC-NLSTM | 0.976 | 0.968 | 0.961 | 0.972 | 9.544 | 1.32 | 1.181 | 1.219 | 1.231 | 0.456 | 0.0028 | 0.0104 | 0.0012 | 0.0008 | 0.0011 |
| Proposed | 0.985 | 0.975 | 0.967 | 0.967 | 9.645 | 0.634 | 1.091 | 1.120 | 0.977 | 0.453 | 0.0004 | 0.0005 | 0.0009 | 0.0005 | 0.0004 |

The PIAW metric of our proposed model is the lowest among all tests, with reductions of 48.72%, 51.02%, 44.72%, and 53.34% compared to the QRLSTM, QRGRU, QRNLSTM, and QRCNN-LSTM models, respectively. This is due to the pre-processing of the data by SWT. SWT-QRADSC-NLSTM has a 13.38% reduction in PIAW compared to SWT-QRNLSTM, which indicates that the improvement in interval prediction performance depends not only on the quadratic decomposition of the SWT but is also determined by the structure of

our feature extraction module ADSC-NLSTM. NLSTM can effectively solve the temporal. The combination of the long-term dependence problem, which is effectively addressed by the ADSC module with an attention mechanism that focuses on dynamic features in the power data, improves the prediction accuracy and generalization of the model. Comparing SWT-QRADSC-NLSTM with the proposed model, the PIAW of SWT-QRTCN-ADSC-NLSTM decreases by 16.32% on average. This demonstrates the effectiveness of this paper's approach of using TCN and ADSC-NLSTM to predict the high and low-frequency components separately and extract different frequency features.

The interval prediction results of different models are shown in Figure 9. The trend of the SWT-QR TCN-ADSC-NLSTM model's upper and lower bounds coincides with the actual value's retention. The prediction interval obtained by the model basically covers all the actual values, and its width is narrow. The model consistently maintains an interval width that performs well for the entire forecast dataset, regardless of whether the power variation is smooth or dramatic. In addition, the model has the narrowest prediction interval at the highest coverage, indicating that we can guarantee the highest interval sensitivity while still meeting reliability requirements.

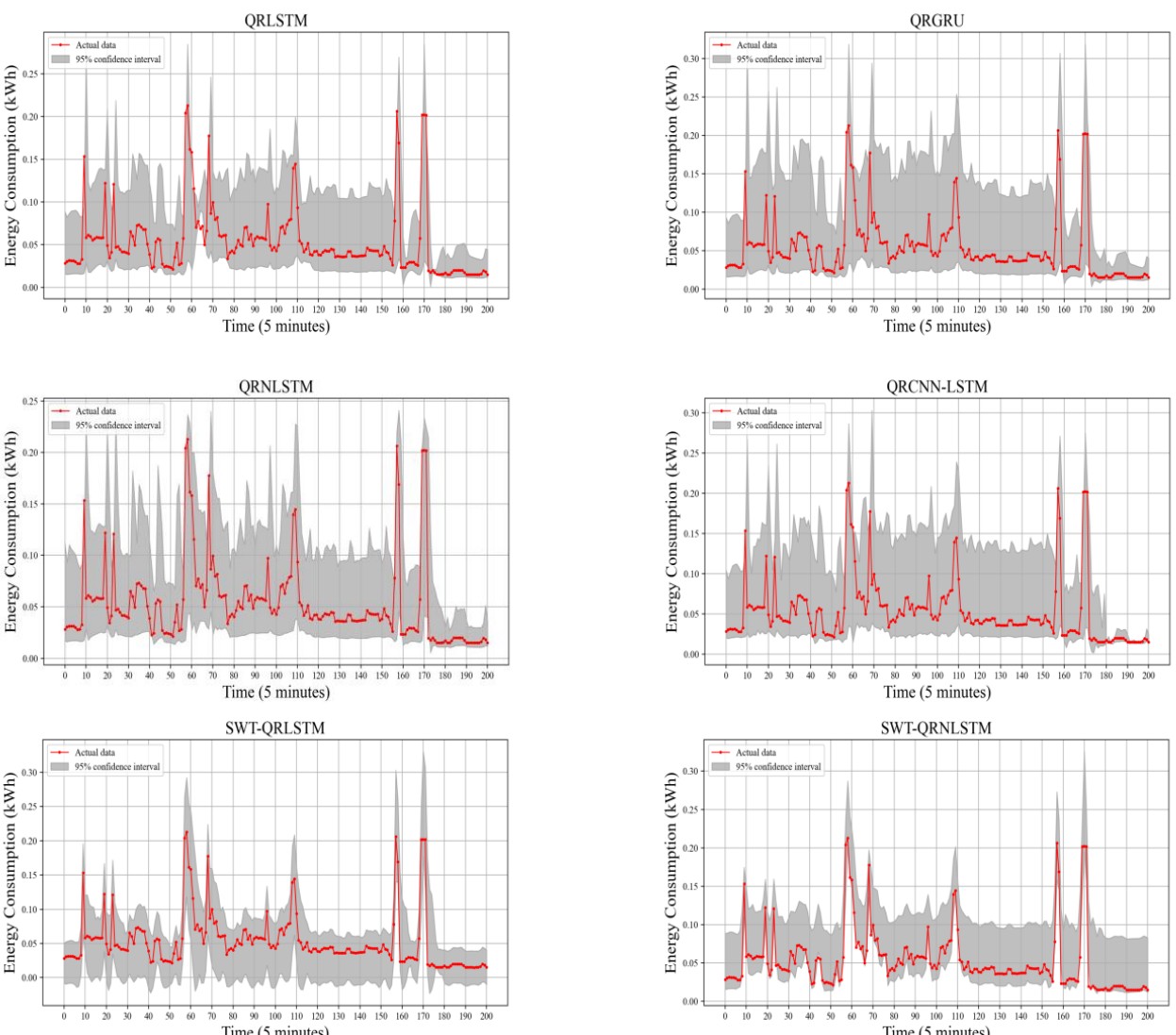

**Figure 9.** *Cont.*

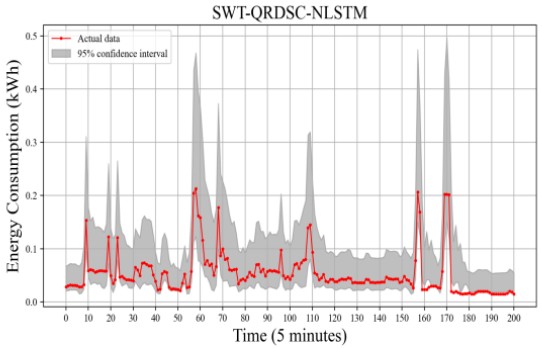
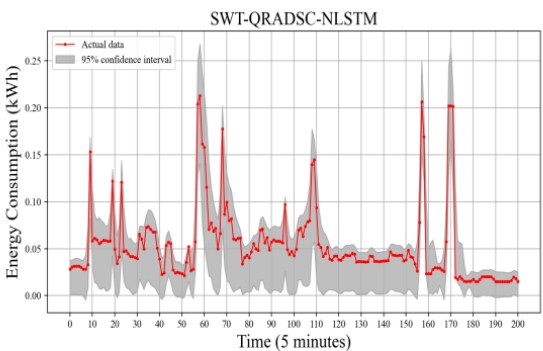
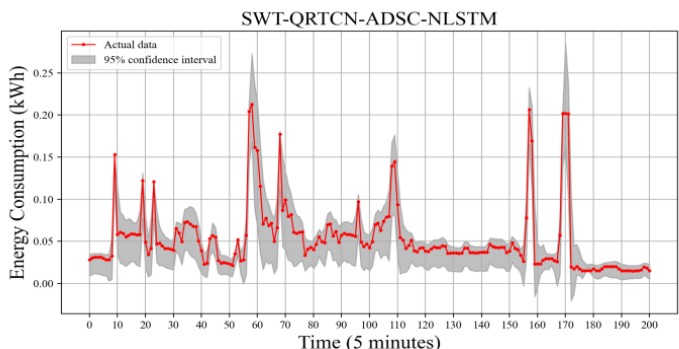

**Figure 9.** Interval prediction results of different models.

### 4.3.3. Probability Prediction Results Analysis

This paper selects the three-time points with the largest data volatility in Figure 8: the 58th, 68th, and 157th, respectively. We also selected three relatively gentle time points: the 31st, 96th, and 126th. The shapes of the probability density curves of the six moments, as mentioned above, are shown in Figure 10. All probability density curves are round and full, with no spikes or dips. The highest peaks are less different, not too wide, and not too narrow, indicating that the probability density curves are appropriate. Most of the observations are close to the highest point of the probability density, and a few observations are off-center, indicating that the probability predictions are reliable. The actual values of electricity consumption are close to the peak of the PDF curves, which shows that this model can effectively perform the probabilistic prediction of electricity consumption. The CRPS metric values for the probabilistic prediction results of the different models are shown in Table 2. The CRPS of the model proposed in this paper is the smallest in all five household datasets, indicating that this model has the highest comprehensive probabilistic prediction performance.

In graphical form, we present the results and evaluation metrics for point prediction, interval prediction, and probabilistic prediction: (1) There is a significant improvement in point prediction accuracy compared to the cutting-edge DL method. The MAPE values have been reduced to below 4% for all five different patterns of household energy consumption forecasts. (2) Most of the actual values fall within the high confidence level prediction interval, reflecting the validity of the model interval prediction proposed in this paper. (3) The true values in the probability density graph are almost always distributed at the higher probability density values, indicating that the prediction results of this paper's method are highly referable.

In summary, the electricity consumption forecasting model exhibits very good performance in both point and probabilistic forecasting. Our model can obtain households' electricity demand in future periods accurately and in real-time, providing people with decision support and helping them adjust their household equipment usage time.

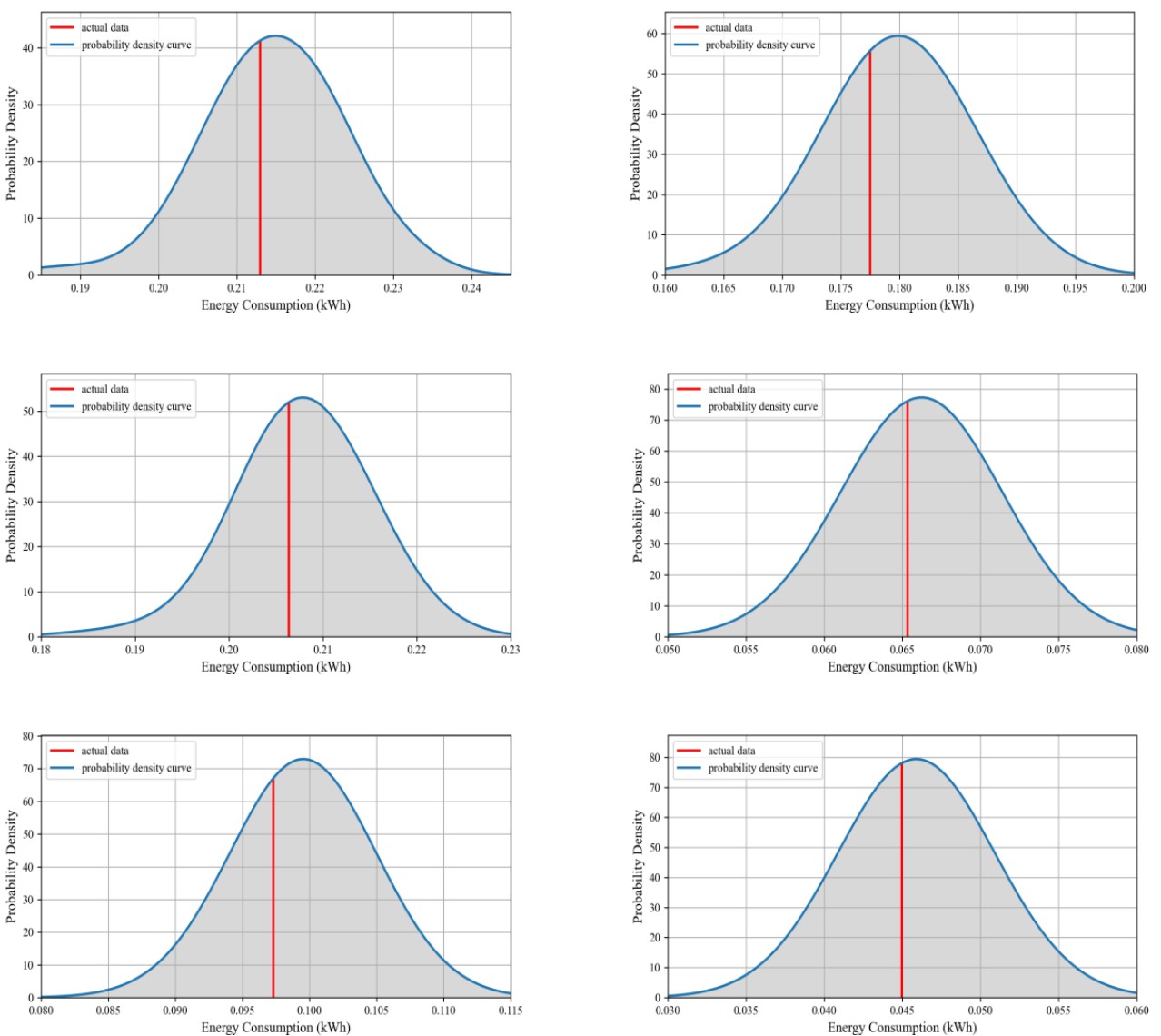

**Figure 10.** Probability density curves at different time points predicted by SWT-QRTCN-ADSC-NLSTM.

## 5. Conclusions

Residential energy consumption forecasting is essential for the electricity market and individual households. Researchers have been striving to implement new methods to improve prediction accuracy. SWT is a more reliable decomposition method, TCN can extract multi-scale features, BiNLSTM can obtain time-series dependencies, DSC can better learn the useful features, and CBAM can focus on the critical information of the model. These methods have good performance in power forecasting. QR is a reliable method commonly used in interval forecasting.

Therefore, a new hybrid model SWT-QRTCN-ADSC-NLSTM is constructed in this paper to forecast household energy consumption. The DL model's multi-channel input data is constructed using the SWT method, which reduces the complexity of the data. Relatively smooth low-frequency subsequences are used as inputs to the TCN. High-frequency subseries containing sharp fluctuations are used as input to the ADSC-NLSTM, which combines the strengths of each model to improve the prediction accuracy for high-frequency series. Finally, the DL module is combined with QR and KDE to achieve interval prediction and probabilistic prediction.

It is shown experimentally that the proposed model achieves highly reliable and sensitive interval prediction and effective probabilistic prediction while ensuring the accuracy

of point prediction. Our proposed forecasting framework has important implications for reducing energy costs, improving energy efficiency, and increasing living comfort.

The multi-channel input data of this model was constructed using SWT, which reduces the complexity of the data and improves the predictive power of the model.

**Author Contributions:** Conceptualization, K.Y.; methodology, K.Y.; software, L.S.; validation, K.Y.; formal analysis, K.Y.; investigation, N.J.; resources, N.J.; data curation, N.J.; writing—original draft preparation, L.S.; writing—review and editing, G.J.H. and K.Y.; visualization, L.S.; supervision, K.Y. and G.J.H.; project administration, K.Y. and G.J.H.; funding acquisition, K.Y. and G.J.H. All authors have read and agreed to the published version of the manuscript.

**Funding:** This work was supported by Singapore MOE AcRF Tier 1 fundings with grant numbers A-0008299-00-00 and A-0008552-01-00 (K.Y.).

**Data Availability Statement:** Data used is available upon request.

**Conflicts of Interest:** The authors declare no conflict of interest.

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
