# Peer review of "Probabilistic Forecasting of Residential Energy Consumption Based on SWT-QRTCN-ADSC-NLSTM Model"

_information, doi:10.3390/info14040231_

Round 1

Reviewer 1 Report

This paper describes an approach for bot point and probabilistic forecasting of residential energy consumption based on a combination of multiple techniques. The work described is generally interesting, but there are several issues to be handled.

First, a full review of the English language should be performed, including the rules concerning the punctuation marks (e.g. include a space after each punctuation mark).

A deeper justification of the proposed approach would be fundamental to understand the rational of applying such a complex methodology to data with such strong stochastic and noisy (unpredictable) characteristics. Moreover, several methodological options are not fully justified: why using a TCN for low-frequency components and ADSC-NLSTM for the high-frequency ones?

Form lines 296-297, I assume the forecasting horizon is 5 min, i.e., authors the goal is to perform load forecasting 5-min ahead. It should be mentioned in the Abstract and in the Introduction.

It would be quite interesting to include some words about a possible application for a 5-min ahead household load forecasting. Note that, in power systems operation, 5min is not enough to take any corrective actions. Besides, the most important is not the load of a single house but the aggregated value of a large set of them in the same network. However, applying this (rather complex) approach to, for example, 400 households could be quite demanding.

The results seems very good for the tested cases. However, I'm not convinced these results can be generalized. The large peaks of an individual consumers depend a lot on local circumstances and inhabitants’ decisions: how could we predict a given inhabitant will turn on the water kettle in the next 5 min? or that she/he will start cooking? Despite the consumer habits, this kind of events are basically unpredictable and that is the reason of my previous comment on the applicability of such complex system to a problem that does not seem inherently complex.

If I interpreted correctly, the trained system is able to predict the load of the 5 households, but it will probably fail for other cases outside this sample.

Other details:

Some references (e.g. [1], [2]) seem not the most adequate to support the sentences where they are included.

Article [4] is a survey on solar irradiance forecasting; not on household load forecasting.

In line 49, Check if it is really the reference [6] that you meant, and not [5].

The article [9] does not present a structured classification of forecasting methods as suggested by the sentence in lines 89-91.

A few acronyms are defined in the Introduction, although they already have been defined in the Abstract.  It is not usual to redefine acronyms but check the Journal rules. 

Th acronym ADSC-NLSTM (line 62) was not defined.

The last bullet of the contribution (page 2, line 69) seems not a specific contribution of this paper. It basically points out the main advantages of probabilistic forecasting compared to point forecasting.

Check sentence in lines 96-98, namely the part “such as Wei,ZZ Wei et al. [12] proposed”.

Chapter 2 (Related Works) refers interesting research articles made by other authors. However, it is not clear why some of them are referred (just for a matter of example?).  Most often, the main goal of the bibliographic review is to enhance what the novelty of the proposed approach.

Line 217 – Authors neither explain the concept of "Inflated causal convolution", nor present a reference to it.

With this regard, I only found 2 references on this topic. One of then has a figure that is very similar to the left part of Figure 4. See the following reference:

Lv X, Xiong X and Geng B (2023), Increasing the prediction performance of temporal convolution network using multimodal combination input: Evidence from the study on exchange rates. Front. Phys. 10:1008445. doi: 10.3389/fphy.2022.1008445

Lines 243-244 – The text says “DSC is combined with CBAM, where SAE allows the DSC network to focus more on the mutation regions in the data sequence”. –  Which "mutation regions"? Could you exemplify? Do authors mean, that beyond the calendar, temperature and other exogenous effects, there are other mutation regions, not explained by those factors?

Author Response

Reviewer 1:

This paper describes an approach for bot point and probabilistic forecasting of residential energy consumption based on a combination of multiple techniques. The work described is generally interesting, but there are several issues to be handled.

First, a full review of the English language should be performed, including the rules concerning the punctuation marks (e.g. include a space after each punctuation mark).

Reply: Thanks for the reviewer’s comments. We have carefully corrected the language errors and asked several native English speakers to proofread the papers. We believe that English writing has improved tremendously. At the same time, we have checked and revised formatting issues throughout the article including punctuation rules, sentence case, and common errors.

It is noted that minor English editions may not be reflected in red color in the original paper.

A deeper justification of the proposed approach would be fundamental to understand the rational of applying such a complex methodology to data with such strong stochastic and noisy (unpredictable) characteristics. Moreover, several methodological options are not fully justified: why using a TCN for low-frequency components and ADSC-NLSTM for the high-frequency ones?

Reply: The use of complex network models can often be better adapted to changes in energy consumption data with strong stochastic characteristics, improving the robustness and generalization of the model.

SWT can decompose the data into high and low frequency sub-signals, representing the fast-changing part and the gently changing part of the original timing signal, respectively. Data decomposition helps subsequent models better capture different patterns and trends in time series signals.

A parallel network model consisting of NLSTM, DSC, and CBAM can better handle prediction tasks for data with strong randomness characteristics. NLSTM, an improved LSTM model, can better handle data with randomness and noise, helping the model to better capture the long-term dependencies of the data. And DSC combined with the CBAM attention mechanism can help the model better identify and exploit important information and features in the data. By combining these models, their respective strengths can be leveraged to improve the performance of the model.

We have a rough explanation of using a TCN for low-frequency components and ADSC-NLSTM for the high-frequency ones in the introduction section. In the revised version,we have added  more detailed descriptions in 3.5 Combination model of electricity consumption probability prediction.

“According to the different frequency characteristics of the decomposed subsequence, selecting different DL models for data training can better capture the different change patterns and trends of temporal signals.The low-frequency main signal becomes smooth after the removal of high-frequency noise, and a normal single model is able to capture its long-term trend. If complex models are used with too many training parameters, they not only increase the training time but may also lead to over-fitting.TCN can effectively grasp the change law of stationary signal in timing under the case of few model parameters. This method is applied to predict low frequency components.In contrast, the high-frequency signal obtained from the decomposition still contains both frequent and complex fluctuations, and it is difficult for a single model to capture the complex non-linear features. Therefore, a combined model with high robustness, ADSC-NLSTM, is proposed and applied to the high frequency section.”

Form lines 296-297, I assume the forecasting horizon is 5 min, i.e., authors the goal is to perform load forecasting 5-min ahead. It should be mentioned in the Abstract and in the Introduction.

It would be quite interesting to include some words about a possible application for a 5-min ahead household load forecasting. Note that, in power systems operation, 5min is not enough to take any corrective actions. Besides, the most important is not the load of a single house but the aggregated value of a large set of them in the same network. However, applying this (rather complex) approach to, for example, 400 households could be quite demanding.

Reply: Thanks for the reviewer’s comments. In the revised version, we have rearranged the content of the abstract and the introduction.

We reworded the abstract in line with the recommendations:

“The model was tested using historical demand-side data from 5 UK households to achieve energy consumption predictions 5 minutes in advance”

We added an applicable description of 5-min ahead household load forecasting in the Introduction.

“By realizing ultra-short-term home power prediction in 5-minute steps, understanding the real-time electricity consumption of households can help people respond to emergency power accidents in time. At the same time, it can better balance the power load, and improve the efficiency of real-time dispatch and energy management of renewable energy resources in the microgrid[4].With a large number of smart homes in use, accurate forecasting of energy consumption demand has become an important part of deploying an energy management system (EMS)[5].Consumers can monitor their electricity usage in real time and take some energy-saving measures to reduce energy bills.”

Reference:

  1. Marzband M, Ghazimirsaeid S S, Uppal H, et al. A real-time evaluation of energy management systems for smart hybrid home Microgrids[J]. Electric Power Systems Research, 2017, 143: 624-633.
  2. El-Baz W, Tzscheutschler P. Short-term smart learning electrical load prediction algorithm for home energy management systems[J]. Applied Energy, 2015, 147: 10-19.

The electricity data used in this paper is a publicly available dataset provided by the UK Energy Centre [1], which contains only five household electricity datasets. There have been many researchers who have carried out research on prediction of household electricity for this purpose. For the energy consumption prediction problem of individual households, Yan K et al.[2] proposed a hybrid integrated deep learning framework, noting that using wavelet decomposition for data pre-processing can effectively improve the prediction accuracy of LSTM neural networks. Jin N et al.[3] proposed a hybrid model combining singular spectrum analysis and LSTM, which can extract hidden features from the raw data and capture long-term dependencies to achieve high-precision energy consumption prediction for each household.

Reference:

[1] J. Kelly, W. Knottenbelt, The UK-DALE dataset, domestic appliance-level electricity demand and whole-house demand from five UK homes, Sci. Data 2 (1) (2015)1–14.

[2] Yan K, Li W, Ji Z, et al. A hybrid LSTM neural network for energy consumption forecasting of individual households[J]. Ieee Access, 2019, 7: 157633-157642.

[3] Jin N, Yang F, Mo Y, et al. Highly accurate energy consumption forecasting model based on parallel LSTM neural networks[J]. Advanced Engineering Informatics, 2022, 51: 101442.

Overall power forecasts for a large number of homes do provide a better understanding of overall trends in market and user demand to better support decision making.Due to the limited data set in this paper, the energy consumption prediction of a large number of houses cannot be temporarily realized. In the later stage, we will use more extensive data of real world energy consumption data to carry out the power prediction research on a large number of houses and longer time intervals.

All modifications of the newer version of the manuscript are marked in red color to the reviewer’s easy reference.

The results seems very good for the tested cases. However, I'm not convinced these results can be generalized. The large peaks of an individual consumers depend a lot on local circumstances and inhabitants’ decisions: how could we predict a given inhabitant will turn on the water kettle in the next 5 min? or that she/he will start cooking? Despite the consumer habits, this kind of events are basically unpredictable and that is the reason of my previous comment on the applicability of such complex system to a problem that does not seem inherently complex.

Reply: Thanks for the reviewer’s comments.

First , it must be recognized that the suitability of the model to the sample data is an unavoidable problem in the field of time series forecasting, and almost all time series forecasting relies fully on historical data distribution. In the field of home electricity forecasting, the proposed method has significantly better prediction effect than other methods. Then, in other similar problem research, I believe that there will be not-bad results.

Second, individual household electricity data is more random, unstable, and volatile compared to larger commercial buildings and regional total electricity loads. This is mainly due to the different composition of individual family members and the unpredictability of members' behavior. In addition, energy consumption is also affected by the season and weather, which increases the challenge for the training system. This has always been a difficult point and challenge in time series forecasting.

Finally, with the development of smart grids and the modernization of measurement and control systems, flexible scheduling and resource management on the user side are gradually improving.More and more companies are launching their own smart home energy management systems. These measures reflect the importance of household electricity loads and illustrate the inevitability of forecasting from a macro to a micro perspective.

Other details:

Some references (e.g. [1], [2]) seem not the most adequate to support the sentences where they are included.

Reply: Thanks for the reviewer’s comments. In the revised edition, we reviewed the cited references cited, made changes to the inappropriate literature and added some discussion to ensure that our paper was convincing enough. All modifications of the newer version of the manuscript are marked in red color to the reviewer’s easy reference.

Article [4] is a survey on solar irradiance forecasting; not on household load forecasting.

Reply: Thanks for the reviewer’s comments. In the revised edition, This Article has been removed.

In line 49, Check if it is really the reference [6] that you meant, and not [5].

Reply: Thanks for the reviewer’s comments. We have modified the literature [6] in the original manuscript, which is shown as literature [9] in the revised version.

Reference:

  1. Ye Z, O’Neill Z, Hu F. Hardware-based emulator with deep learning model for building energy control and prediction based on occupancy sensors’ data[J]. Information, 2021, 12(12): 499.

The article [9] does not present a structured classification of forecasting methods as suggested by the sentence in lines 89-91.

Reply: Thanks for the reviewer’s comments. We have modified the literature [9] in the original manuscript, which is shown as literature [10] in the revised version.

References:

  1. Moradzadeh A, Mansour-Saatloo A, Nazari-Heris M, et al. Introduction and literature review of the application of machine learning/deep learning to load forecasting in power system[J]. Application of Machine Learning and Deep Learning Methods to Power System Problems, 2021: 119-135.

A few acronyms are defined in the Introduction, although they already have been defined in the Abstract.  It is not usual to redefine acronyms but check the Journal rules. 

Th acronym ADSC-NLSTM (line 62) was not defined.

 Reply: Thanks for the reviewer’s comments.I deleted the repeat definition initials and added the definition of ADSC-NLSTM.

 “In this study, a hybrid model combining BiNLSTM, DSC and attention, namely ADSC-NLSTM, is constructed for the prediction of high frequency sequences.”

The last bullet of the contribution (page 2, line 69) seems not a specific contribution of this paper. It basically points out the main advantages of probabilistic forecasting compared to point forecasting.

Reply: Thanks for the reviewer’s comments. In the revision, the last bullet of the contribution was changed.

“This model is compared with existing frontier technologies including individual models and combined models, and the experimental results show that this model achieves efficient energy consumption point prediction and probability prediction.”

Check sentence in lines 96-98, namely the part “such as Wei,ZZ Wei et al. [12] proposed”.

Reply: Thanks for the reviewer’s comments.In the revision, “Wei, ZZ Wei et al.” has been modified to “such as Wei R R et al.”

  • Chapter 2 (Related Works) refers interesting research articles made by other authors. However, it is not clear why some of them are referred (just for a matter of example?).  Most often, the main goal of the bibliographic review is to enhance what the novelty of the proposed approach.

Reply: Thanks for the reviewer’s comments. We list the current methods used in timing prediction and point out their existing problems. In the literature review, we listed several enlightening articles for this experiment, for example, literature [29], [30]. In addition, we studied the current literature using quantile regression for probability interval prediction. 

(6)Line 217 – Authors neither explain the concept of "Inflated causal convolution", nor present a reference to it.

With this regard, I only found 2 references on this topic. One of then has a figure that is very similar to the left part of Figure 4. See the following reference:

Lv X, Xiong X and Geng B (2023), Increasing the prediction performance of temporal convolution network using multimodal combination input: Evidence from the study on exchange rates. Front. Phys. 10:1008445. doi: 10.3389/fphy.2022.1008445

Reply: Thanks for the reviewer’s comments. "Inflated causal convolution" and "dilated causal convolution" are the same, only with different expressions.I am very sorry for my negligence, which caused the confusion to the reviewer. For the consistency of the context, we uniformly express it as "dilated causal convolution".

In the revised version,More detailed description has been added:

“Dilated causal convolution consists of causal convolution and dilated convolution[36]. Causal convolution is a strict time-constrained model that prevents future data from leaking into past data. Dilated convolution samples input at intervals on the basis of causal convolution. It adjusts the size of the receptive field by changing the expansion coefficient, which enables the network to flexibly adjust the amount of historical information received by the output. The residual links in the residual block enable the network to pass information across the number of layers, thus avoiding the loss of information caused by too many layers.”

  • Lines 243-244 – The text says “DSC is combined with CBAM, where SAE allows the DSC network to focus more on the mutation regions in the data sequence”. –  Which "mutation regions"? Could you exemplify? Do authors mean, that beyond the calendar, temperature and other exogenous effects, there are other mutation regions, not explained by those factors?

Reply: Thanks for the reviewer’s comments.The "mutation regions" refers to the peak part in the data sequence. The dataset used in this study contains only energy consumption data and no other exogenous effects.

Reviewer 2 Report

After reading the paper i can say that the paper is good and contains scientific contribution 

but some comments should be done before accepting the paper for publication

1- The graph is good but needs some elaboration 

2- Also what is the main contribution of the paper??

3-  What is the main use of equation 1 and where it can be replaced or not?

4-Equation 25 should be checked

5- The literature needs to be organized.

6- The authors must add concluded remarks from the Experimental Process and Results section.

Author Response

Reviewer 2:

After reading the paper i can say that the paper is good and contains scientific contribution but some comments should be done before accepting the paper for publication

  • The graph is good but needs some elaboration 

Reply: Thanks for the reviewer’s comments.We have carefully examined the inadequacy of the graph description in the paper and added some elaboration to the graph.

We have added instructions to the subgraphs in Figure 2,4,6 and put them in their respective titles

Figure 2. The decomposition component after wavelet transform.The graphs on the right represents the raw data from top to bottom, A3, D3, D2, A3, D1-A1 and D1-D1. Only 600 data samples were depicted for better and clearer vision.

Figure 4. The structure of TCN. The graph on the right represents the dilated causal convolution, and the left shows the residual block.

Figure 6.The structure of CBAM.The middle part of the figure is the overall structure of CABM, the top is the internal structure of CAM, and the bottom is the internal structure of SAM.

We have added a more detailed explanation of the TCN structure in Figure 4 and he CBAM structure in Figure 6.

In addition, we statistically formulated the graphs and tables of all the work results.

All modifications of the newer version of the manuscript are marked in red color to the reviewer’s easy reference.

  • Also what is the main contribution of the paper??

Reply: Thanks for the reviewer’s comments.

We have revised the introduction section making the research gap, goal and contribution clearer. Specifically, the objective of the paper is proposing a novel energy consumption prediction algorithm that outperforms the existing methods in terms of accuracy, reliability and availability. And we stated the contributions clearly at the end of the introduction section.

“•  The original data is decomposed by smooth wavelet decomposition to generate multiple sub-signals, and its high-frequency signal is decomposed twice and trained using TCN and ADSC-NLSTM for both low and high frequency signals, and this combination helps to solve the model adaptation problem for electricity consumption prediction. 

  • The proposed ADSC-NLSTM network is capable of making predictions in space and time, where DSC based on the attention mechanism can preserve important information backwards, prevent information loss and enhance the performance of feature extraction. The nested structure of the BiNLSTM is used to effectively perform deep feature extraction, and there are more recurrent units to obtain the dependencies of features at each time point, allowing the network to learn more adequately.
  • This model is compared with existing frontier technologies including individual models and combined models, and the experimental results show that this model achieves efficient energy consumption point prediction and probability prediction.”

In a few words,we combine different methods to fully exploit the strengths of each model in order to build a highly robust forecasting model. Highly accurate prediction results are achieved for the electricity consumption of different households

All modifications of the newer version of the manuscript are marked in red color to the reviewer’s easy reference.

  • What is the main use of equation 1 and where it can be replaced or not?

Reply: Thanks for the reviewer’s comments. Equation 1 is used to explain the wavelet decomposition process. After the first decomposition, the original data is divided into low-frequency subsequence A1 and high-frequency subsequence D1. A1 is used as the object of the next decomposition, giving A2 and D1, D2. In this paper, the wavelet transform is set to three times to decompose the original data Y(t) to obtain four subsequences: A3, D1, D2, and D3. Then the complex D1 component is again based on Equation 1 to obtain D1-A1, D1-D1.

Equation 1 in the article is a general representation of the wavelet transform and can be replaced with a more detailed formula:

where, H represents the high pass filter and L represents the low pass filter, after each decomposition, the original data is divided into low frequency subsequence A and high frequency subsequence D, where the low frequency subsequence A is taken as the object of the next decomposition.The meaning of the two formulas is the same, and the equation 1 in our paper are commonly used, so we do not recommend substitution.

  • Equation 25 should be checked

Reply: Thanks for the reviewer’s comment.

I checked Equation 25 and found a missing definition of I. In the revision, I have added a definition of I

I is Heaviside function”

  • The literature needs to be organized.

Reply: Thanks for the reviewer’s comment.

We have restructured the literature review section to make it more logical.

First, I present the problems and challenges faced in studying the field.

Then. I start with three types of currently popular models for temporal prediction. The popular approaches based on traditional models, machine learning and deep learning are listed respectively, and the possible problems with traditional approaches and machine learning are pointed out.

Next, I point out the limitations of a single deep learning model to elicit the effectiveness of data decomposition for prediction. We show the work of several researchers and the insightfulness of this work,

Finally,present the method we use for probabilistic prediction - quantile regression, and demonstrate the effectiveness of quantile regression in probability through citations in the literature

  • The authors must add concluded remarks from the Experimental Process and Results section.

Reply: Thanks for the reviewer’s comment.

In the Experimental Process and Results section.,We have discussed the results of point prediction, interval prediction and probability prediction in detail, including the statistical calculation of the data table and the visual description of the graph

Certainly,I have also made a conclusive summary of the three predictions:

“We present the results and evaluation metrics for point prediction, interval prediction and probabilistic prediction in graphical form. Firstly, there is a significant improvement in point prediction accuracy compared to the cutting edge DL method. The MAPE values have been reduced to below 4% for all five different patterns of household energy consumption forecasts. Secondly, most of the actual values fall within the high confidence level prediction interval, reflecting the validity of the model interval prediction proposed in this paper. Finally, the true values in the probability density graph are almost always distributed at the higher probability density values, indicating that the prediction results of this paper's method are highly referable.

In summary, the electricity consumption forecasting model exhibits very good forecasting performance in both point forecasting and probabilistic forecasting. Our model is able to obtain the electricity demand of households in future time periods accurately and in real time, providing people with decision support and helping them to adjust their household equipment usage time.”

All modifications of the newer version of the manuscript are marked in red color to the reviewer’s easy reference.

Reviewer 3 Report

In the current period marked by the COVID-19 crisis and the Russian-Ukrainian war, energy prices and consumption have become increasingly volatile, being strongly affected, so making an efficient forecast is a real challenge. Thus, I consider the topic addressed to be interesting.

But, the paper must be improved in some sections:

- In the Introduction, the authors must highlight a synthesis of the results obtained and the structure of the work;

- A review of the literature should be described in more detail which is poor in its current form;

- The methodology section must contain bibliographic references;

- In line 168 does reference 202 appear? Where from? The paper has 29 references...;

- The results must be discussed and compared with the results of other studies... it is necessary that the authors must draw conclusions that be sustained by evidence from other research;

- The paper's final section could discuss the most important theoretical and practical implications emerging from the study, the possibility of generalizing the study results, and present possible limits of the current study. At the same time, the authors can point out what the future research directions are;

- It is necessary to include more practical implications of your results that can help academic communities and policymakers. This will help you emphasize the importance of your study compared to similar ones.

Author Response

Reviewer 3:

In the current period marked by the COVID-19 crisis and the Russian-Ukrainian war, energy prices and consumption have become increasingly volatile, being strongly affected, so making an efficient forecast is a real challenge. Thus, I consider the topic addressed to be interesting.

But, the paper must be improved in some sections:

1-In the Introduction, the authors must highlight a synthesis of the results obtained and the structure of the work;

Reply: Thanks for the reviewer’s comment.

A synthesis of the results has been mentioned in the last point of the main contribution: “This model is compared with existing frontier technologies including individual models and combined models, and the experimental results show that this model achieves efficient energy consumption point prediction and probability prediction.

2-A review of the literature should be described in more detail which is poor in its current form;

Reply: Thanks for the reviewer’s comment.

In the literature review section, we discuss the literature of is interest and enlightening.

“Cai et al. [27] decomposed the power load sequence data into modal components of different frequencies by VMD. They used GRU and TCN to predict the low-frequency and high-frequency components, respectively. Inspired by this, we make predictions for the components of different frequencies using the corresponding model. Shao et al. [28] used discrete wavelet transform (DWT) reconstructed sequences as the input for power consumption prediction. Then they built a combined prediction model based on CNN-LSTM for each subseries. However, like most wavelet decomposition methods, this article only focuses on the decomposition of low-frequency approximate signals, not high-frequency signals. In view of this problem, Zeng Y et al. [29] proposed a new model combining ESWT and NSTM and performed well in the air quality prediction effect. Liang Y.et al.[30] combined ICEEMDAN, LSTM, CNN and CBAM to create a collaboration that has built a powerful model. This joint approach can take full advantage of the respective algorithms' strengths to improve the model's performance and robustness. However, the ICEEMDAN algorithm has a long computation time, which is not conducive to short-term prediction.”

3-The methodology section must contain bibliographic references

Reply: Thanks for the reviewer’s comment.We haved added the literature on methods used, including:

34  Nason G P, Silverman B W. The stationary wavelet transform and some          statistical      applications[J]. Wavelets and statistics, 1995: 281-299.           

35   Moniz J R A, Krueger D. Nested lstms[C]//Asian Conference on Machine Learning. PMLR, 2017: 530-544

36   Mishra K, Basu S, Maulik U. DaNSe: a dilated causal convolutional network based model for load forecasting[C]//Pattern Recognition and Machine Intelligence: 8th International Conference, PReMI 2019, Tezpur, India, December 17-20, 2019, Proceedings, Part I. Springer International Publishing, 2019: 234-241.

37   Guo M H, Xu T X, Liu J J, et al. Attention mechanisms in computer vision: A survey[J]. Computational Visual Media, 2022, 8(3): 331-368.

38  Koenker R, Hallock K F. Quantile regression[J]. Journal of economic perspectives, 2001, 15(4): 143-156.

39   Węglarczyk S. Kernel density estimation and its application[C]//ITM Web of Conferences. EDP Sciences, 2018, 23: 00037.

40  Xu X, Yan Z, Xu S. Estimating wind speed probability distribution by diffusion-based kernel density method[J]. Electric Power Systems Research, 2015, 121: 28-37.

4-In line 168 does reference 202 appear? Where from? The paper has 29 references...;

 Reply: Thanks for the reviewer’s comment. All references have been checked and corrected.

“[200]” has been deleted

5-The results must be discussed and compared with the results of other studies... it is necessary that the authors must draw conclusions that be sustained by evidence from other research;

Reply: Thanks for the reviewer’s comment. In the Experimental Process and Results section.

we have performed a comparative analysis of the proposed model and the contrast model.

The point prediction results were analyzed as follows:

”The results of the point prediction evaluation metrics for the various models in the Family 1 test set are shown in Table 1, where the MAE, RMSE and MAPE of the proposed algorithm are significantly reduced compared to the comparison models.

SWT-QRDSC-NLST, SWT-QRADSC-NLSTM and our proposed method were decomposed quadratically. Compared to SWT-QRLSTM and SWT-QRNLSTM, the errors of the prediction results of those former models are all reduced to different degrees.The decline is due to the fact that we re-decompose the D1 subseries of the most violent fluctuations. Adequate treatment of the high-frequency components allows for a more efficient and accurate representation of the time series features within the high-frequency subseries.”

The interval prediction results were analyzed as follows:

“The PIAW metric of our proposed model is the lowest among all tests, with reductions of 48.72%, 51.02%, 44,72% and 53.34% compared to the QRLSTM, QRGRU, QRNLSTM and QRCNN-LSTM models, respectively. This is due to the pre-processing of the data by SWT. SWT-QRADSC-NLSTM has a 13.38% reduction in PIAW compared to SWT-QRNLSTM, which indicates that the improvement in interval prediction performance depends not only on the quadratic decomposition of the wavelet transform, but is also determined by the structure of our feature extraction module ADSC-NLSTM. NLSTM can effectively solve the temporal The combination of the long-term dependence problem, which is effectively addressed by the ADSC module with an attention mechanism that focuses on dynamic features in the power data, improves the prediction accuracy and generalisation of the model. Comparing SWT-QRADSC-NLSTM with the proposed model, the PIAW of SWT-QRTCN-ADSC-NLSTM decreases by 16.32% on average. This demonstrates the effectiveness of this paper's approach of using TCN and ADSC-NLSTM to predict the high and low frequency components separately and extract different frequency features.”

The parts of the probability prediction results are analyzed as follows:

“All probability density curves are round and full, with no spikes or dips, and the highest peaks are less different, not too wide and not too narrow, indicating that the probability density curves are appropriate. Most of the observations are close to the highest point of the probability density, and a small number of observations are off-centre, which indicates that the probability predictions are reliable.”

6-It is necessary to include more practical implications of your results that can help academic communities and policymakers. This will help you emphasize the importance of your study compared to similar ones

Reply: Thanks for the reviewer’s comment. In the Methodology section, we elaborate on the principle of the used model introduction and its advantages.

In the Methodology part, we elaborate on the principle of the used model and its advantages. Then we made a detailed analysis of the prediction results in the article, and elaborated our model from all aspects. Finally, in the conclusion section, we summarize the characteristics of each submodel and the specific scheme of the proposed model.

Round 2

Reviewer 2 Report

The paper has been updated , i suggest acceptance 

Reviewer 3 Report

In this version of the updated paper, the authors took into account a part of my observations, and thus I consider their study improved to have.